# Identification and validation of poly-metabolite scores for diets high in ultra-processed food: An observational study and post-hoc randomized controlled crossover-feeding trial

Leila Abar[1], Eurídice Martínez Steele[2,3], Sang Kyu Lee[4], Lisa Kahle[5],
Steven C. Moore[1], Eleanor Watts[1], Caitlin P. O'Connell[1], Charles E. Matthews[1],
Kirsten A. Herrick[6], Kevin D. Hall[7], Lauren E. O'Connor[6], Neal D. Freedman[6],
Rashmi Sinha[1], Hyokyoung G. Hong[4], Erikka Loftfield[1]*

1 Division of Cancer Epidemiology and Genetics, National Cancer Institute (NCI), Rockville, Maryland, United States of America, 2 Department of Nutrition, School of Public Health, University of Sao Paulo, Sao Paulo, Brazil, 3 Center for Epidemiological Studies in Health and Nutrition, University of Sao Paulo, Sao Paulo, Brazil, 4 Division of Cancer Epidemiology and Genetics, Biostatistics Branch, NCI, Bethesda, Maryland, United States of America, 5 Information Management Services, Inc. Beltsville, Maryland, United States of America, 6 Division of Cancer Control and Population Sciences, National Cancer Institute, National Institutes of Health, Rockville, Maryland, United States of America, 7 Laboratory of Biological Modeling, National Institute of Diabetes and Digestive Kidney Diseases, Bethesda, Maryland, United States of America

* erikka.loftfield@nih.gov

## Abstract

### Background

Ultra-processed food (UPF) accounts for a majority of calories consumed in the United States, but the impact on human health remains unclear. We aimed to identify poly-metabolite scores in blood and urine that are predictive of UPF intake.

### Methods and findings

Of the 1,082 Interactive Diet and Activity Tracking in AARP (IDATA) Study (clinical-trials.gov ID NCT03268577) participants, aged 50−74 years, who provided biospecimen consent, *n* = 718 with serially collected blood and urine and one to six 24-h dietary recalls (ASA-24s), collected over 12-months, met eligibility criteria and were included in the metabolomics analysis. Ultra-high performance liquid chromatography with tandem mass spectrometry was used to measure >1,000 serum and urine metabolites. Average daily UPF intake was estimated as percentage energy according to the Nova system. Partial Spearman correlations and Least Absolute Shrinkage and Selection Operator (LASSO) regression were used to estimate UPF-metabolite correlations and build poly-metabolite scores of UPF intake, respectively. Scores were tested in a *post-hoc* analysis of a previously conducted randomized, controlled, crossover-feeding trial (clinicaltrials.gov ID NCT03407053) of 20 subjects who were

**Data availability statement:** IDATA metabolomics data can be shared via a proposal submission and approval process and after an appropriate Data Transfer Agreement is in place. See instructions on access to IDATA data and biospecimens (https://cdas.cancer.gov/learn/idata/instructions/). The trial metabolomics data described in the manuscript will be made available upon request to juen.guo@nih.gov pending completion of a Data Transfer Agreement.

**Funding:** The research leading to these results received funding from the NIH Intramural Research Program at NCI (EL) and NIDDK (KDH). EMS was supported by Fundação de Amparo à Pesquisa do Estado de São Paulo (2023/16144-3). The funders had no role in study design, data collection and analysis, decision to publish, or preparation of the manuscript.

**Competing interests:** I have read the journal's policy and the authors of this manuscript have the following competing interests: LEO is principal investigator on a grant administered by the United States Department of Agriculture's National Institute of Food and Agriculture (USDA-NIFA #2022-07671) to coordinate a workshop to establish a research roadmap for the future of food processing, processed food, and human health research. Collaborators on this grant include scientists from Archer Daniels Midland (ADM) with special input from General Mills and Academic Institutions. All other authors have declared that no competing interests exist.

**Abbreviations:** ASA-24, automated self-administered 24-h dietary assessment tool; AUC, area under the curve; BMI, body mass index; CV, coefficients of variation; ESI, electrospray ionization; FDR, false discovery rate; FMV, first morning void; FNDDS, Food codes were linked to the Food and Nutrient Database for Dietary Studies; ICC, intra-class correlation coefficient; IDATA, Interactive Diet and Activity Tracking in AARP; LASSO, least absolute shrinkage and selection operator; MAE, mean absolute error; NHANES, National Health and Nutrition Examination Survey;

admitted to the NIH Clinical Center and randomized to consume *ad libitum* diets that were 80% or 0% energy from UPF for 2 weeks immediately followed by the alternate diet for 2 weeks; eligible subjects were between 18−50 years old with a body mass index of >18.5 kg/m$^2$ and weight-stable. IDATA participants were 51% female, and 97% completed ≥4 ASA-24s. Mean intake was 50% energy from UPF. UPF intake was correlated with 191 (of 952) serum and 293 (of 1,044) 24-h urine metabolites (FDR-corrected $P$-value < 0.01), including lipid ($n$ = 56 serum, $n$ = 22 24-h urine), amino acid ($n$ = 33, 61), carbohydrate ($n$ = 4, 8), xenobiotic ($n$ = 33, 70), cofactor and vitamin ($n$ = 9, 12), peptide ($n$ = 7, 6), and nucleotide ($n$ = 7, 10) metabolites. Using LASSO regression, 28 serum and 33 24-h urine metabolites were selected as predictors of UPF intake; biospecimen-specific scores were calculated as a linear combination of selected metabolites. Overlapping metabolites included (S)C(S)S-S-Methylcysteine sulfoxide ($r_s$ = −0.23, −0.19), N2,N5-diacetylornithine ($r_s$ = −0.27 for serum, −0.26 for 24-h urine), pentoic acid ($r_s$ = −0.30, −0.32), and N6-carboxymethyllysine ($r_s$ = 0.15, 0.20). Within the cross-over feeding trial, the poly-metabolite scores differed, within individual, between UPF diet phases ($P$-value for paired $t$ test < 0.001). IDATA Study participants were older US adults whose diets may not be reflective of other populations.

## Conclusions

Poly-metabolite scores, developed in IDATA participants with varying diets, are predictive of UPF intake and could advance epidemiological research on UPF and health. Poly-metabolite scores should be evaluated and iteratively improved in populations with a wide range of UPF intake.

## Author summary

### Why was this study done?

- Global production and availability of UPF is high, but accurately measuring UPF consumption is challenging.

- Objectively measured metabolites in blood and urine that derive from the diet or response to dietary intake may be useful for studying UPF intake in epidemiologic studies.

- Complementary observational and experimental human studies provide a unique opportunity to develop and evaluate potential dietary biomarkers.

### What did the researchers do and find?

- Using a discovery metabolomics approach, we identified hundreds of serum and urine metabolites were correlated with percentage energy from UPF intake in 718 free-living adults with diverse dietary intakes who participated in the IDATA Study.

RMSE, root mean squared error; ROC, receiver operator characteristic; SR, standard reference; UPF, ultra-processed food; UPLC, using ultra-high-performance liquid chromatography MS/MS tandem mass spectrometry QC quality control; WWEIA, What We Eat in America.

- We developed and tested poly-metabolite scores that are predictive of UPF intake for blood and urine.

- Using metabolomics data generated previously in a randomized, controlled, crossover-feeding trial of two diets, one high in and the other void of UPF, we evaluated the IDATA poly-metabolite scores. The poly-metabolite scores differentiated, within individual, between the diets that were 80% and 0% energy from UPF.

### What do these findings mean?

- The identified poly-metabolite scores could serve as objective measures of UPF intake in large population studies to complement or reduce reliance on self-reported dietary data.

- Poly-metabolite scores, predictive of UPF intake, could provide novel insight into the role of UPF in human health.

- Limitations: Study participants were older US adults whose diets may vary from other populations. Poly-metabolite scores should be evaluated and iteratively improved in populations with diverse diets and a wide range of UPF intake.

### Introduction

The availability and consumption of ultra-processed food (UPF), defined as ready-to-eat or ready-to-heat, edible, industrially manufactured products made mostly or entirely from ingredients extracted from whole foods and often containing food substances of no or rare culinary use and/or cosmetic additives [1], have increased worldwide [2]. In the United States, UPF accounts for more than 50% of calories consumed by children and adults [3–5] raising concerns about the potential impact on human health. Epidemiological studies have linked UPF consumption to risk factors for chronic disease, such as weight gain and obesity [6–9], as well as to risk of cardiometabolic diseases and certain types of cancer [7,10–12].

Most epidemiological studies on UPF consumption, have used the Nova system to classify food items based on their level of processing [1,12]. Accurate classification according to Nova necessitates detailed information on food sources, processing methods, as well as ingredients and their purposes. However, dietary assessment tools and databases capture this information to varying extents. Consequently, researchers may encounter challenges in accurately applying the Nova system, fueling concerns about exposure misclassification and comparability of study results [13–16].

Previous studies have shown that integrating biomarker data with dietary data, measured via self-report using validated assessment tools, can help to adjust for measurement error and improve the precision of diet-disease association estimates, thereby enhancing statistical power [17,18]. Metabolomics has been used to identify candidate biomarkers for various foods [19], beverages [20], and dietary patterns

[21–23], including a dietary pattern high in UPF [24–26]. Previous studies have reported on cross-sectional UPF-metabolite associations [25,26] and one tightly controlled feeding trial [24] identified hundreds of blood and urine metabolites that differed, within individual, between a dietary pattern high in, compared to one void of, UPF; however, the study had a small sample size and limited menus. Currently, nutritional metabolomics research in population-based studies using validated assessment tools, and serial collection of blood and urine, for generating metabolomics data, is lacking.

The primary aims of this study were to identify serum and urine metabolites associated with average 12-month UPF intake, assessed using multiple 24-h dietary recalls, and to develop blood and urine poly-metabolite scores predictive of UPF intake in a free-living population of participants from the Interactive Diet and Activity Tracking in AARP (IDATA) Study. Our secondary aim was to test whether these poly-metabolite scores could, in the context of the aforementioned, previously conducted, randomized, controlled, crossover-feeding trial, differentiate, within individual, between diets containing 80% and 0% energy from UPF. We hypothesized that diets high in energy from UPF similarly impact the metabolome such that a poly-metabolite score developed in a free-living population will differentiate unprocessed and ultra-processed diet phases in the crossover-feeding trial.

## Materials and methods

### IDATA study design

The IDATA study was designed to evaluate the performance of web-based dietary assessment tools, including the Automated Self-Administered 24-h Dietary Assessment Tool (ASA-24), using reference biomarkers. The IDATA Study design and methods have been described in detail elsewhere [27]. In brief, IDATA participants were recruited from a list of AARP members, aged 50–74 years, who resided in or nearby Pittsburgh, Pennsylvania, spoke English, had internet access, were not on a weight-loss diet, and were free of major medical conditions and mobility limitations. From 2012 to 2013, 1,082 participants were enrolled in the IDATA Study and provided biospecimens consent. The study was approved by the National Cancer Institute Special Studies Institutional Review Board (IRB approval number 11CN155) and is registered on ClinicalTrials.gov (Identifier: NCT03268577); all participants signed informed consent [27].

Our analytic sample included the 718 IDATA participants with dietary data, including at least one ASA-24, and available serum, 24-h urine, and first morning void (FMV) urine, collected at two time points, 6 months apart, for metabolomics analysis (Fig 1). IDATA data can be shared via a proposal submission and approval process and after an appropriate Data Transfer Agreement is in place. See instructions on access to IDATA data and biospecimens at https://cdas.cancer.gov/learn/idata/instructions/.

Participants were divided into four groups for practical study-center related reasons and to reduce the influence of seasonal variation in diet. Each group completed the same collection activities, although the timing varied [28]. Over 12 months, participants completed up to six web-based ASA-24s on a randomly assigned day approximately every other month [27,28]. Each food and beverage item reported was assigned to a unique 8-digit food code based on the foods and beverages reported in What We Eat in America (WWEIA), National Health and Nutrition Examination Survey (NHANES). The structure of the WWEIA dietary data has been described in detail elsewhere [29]. Food codes were linked to the Food and Nutrient Database for Dietary Studies (FNDDS) [30], an application to convert food and beverage portions into gram amounts and to estimate nutrient values, including energy [29,31], using standard reference (SR) codes from the United States Department of Agriculture National Nutrient Database for Standard Reference.

### Nova classification

UPF intake was estimated according to the Nova system, which classifies foods and beverages into one of four groups based on the extent and purpose of industrial food processing. Group 1 items include unprocessed or minimally processed foods such as fresh, dried, or frozen fruits or vegetables, grains, legumes, meat, fish, and milk; group 2 items include processed culinary ingredients which are substances obtained from group 1 foods or from nature such as most

PLOS Medicine

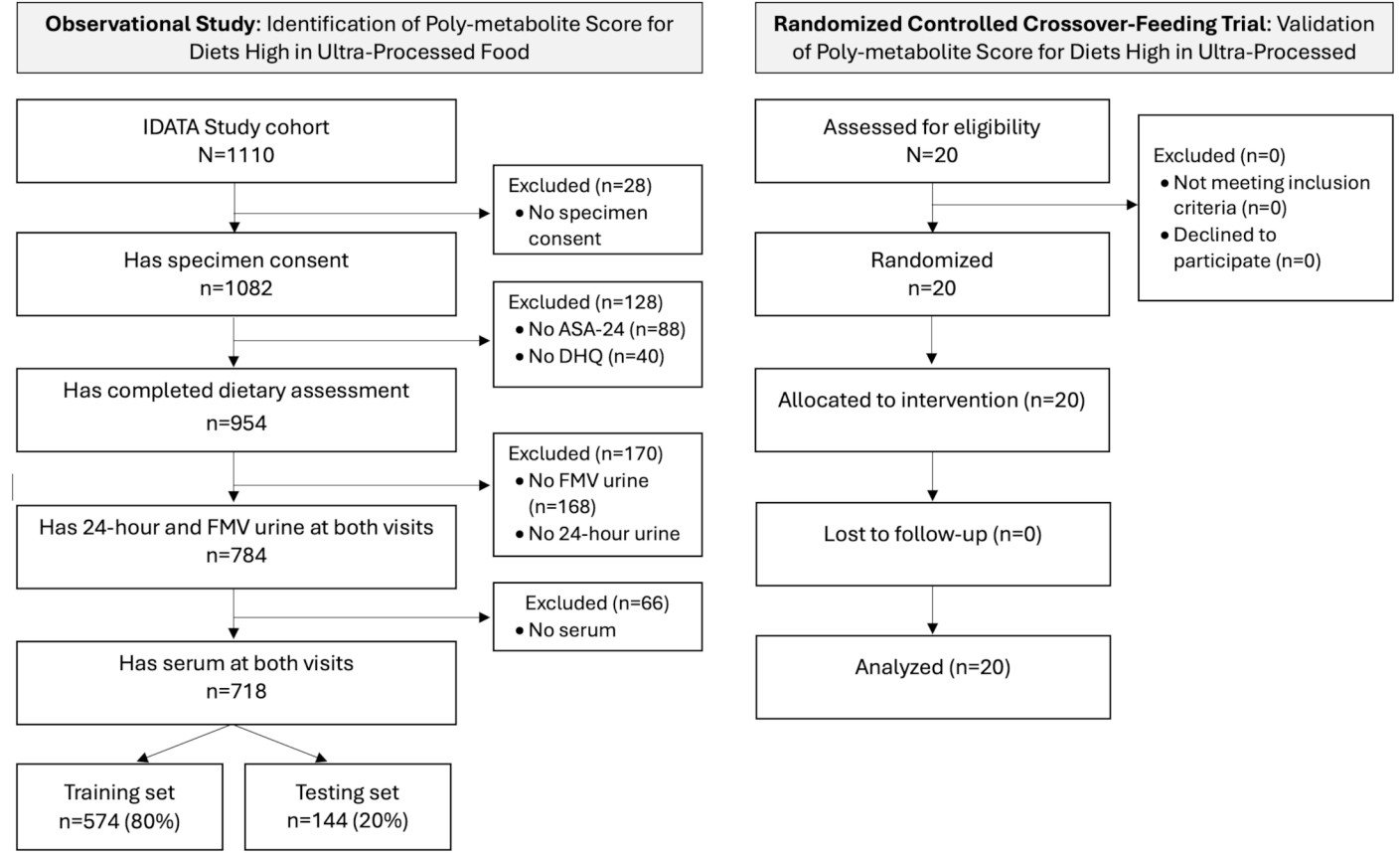

**Fig 1. Participant flowchart and study design.** Abbreviations: ASA-24 = automated self-administered 24-h dietary assessment tool; DHQ = dietary history questionnaire; IDATA = interactive diet and activity tracking in AARP; FMV = first morning void.

plant oils, sugar, honey, and table salt; group 3 items include processed foods, which are made by adding group 2 ingredients to group 1 foods, such as canned vegetables, artisanal breads, and cheese; group 4 items are UPF and include ready-to-eat foods, like industrially manufactured breads and baked goods that contain ingredients not typically used in culinary preparations [1]. Each ASA-24 food and beverage item was disaggregated into FNDDS food and SR codes, which were assigned to 1 of the 4 Nova groups based on the WWEIA, NHANES database developed by Martínez Steele and colleagues [29]. FNDDS food codes ($n = 44$) that had not been previously assigned to a Nova group, were reviewed and assigned a Nova classification using the "reference approach" described by Martínez Steele and colleagues [29]. Next, we calculated total energy intake per day (i.e., total calories) by summing the calories from each of the four Nova groups for each participant per recall day. We then calculated the percentage of total energy intake from UPF (i.e., Nova group 4) for each participant per recall day and averaged percentage of total energy intake from UPF across the number of ASA-24 days completed for each IDATA participant. Average, 12-month percentage of total energy intake from UPF is subsequently referred to as UPF intake. The top 10 food codes contributing to energy intake from UPF accounted for about 12% of energy intake and include ice cream, white rolls, potato chips, and cola-type soft drinks (S1 Table).

## Biological sample collection

Participants visited the study center at months 1, 6, and 12 for assessment of anthropometry measures. During two of the center visits, either months 1 and 6 or months 6 and 12, participants had their blood drawn. Serum samples were kept at −28 °C

and centrifuged within two hours for 15 min at 3,500 rpm prior to being stored at −70 °C. Participants were asked to complete 24-h urine collections at home. Urine collection days occurred approximately 7–10 days after a study center visit. Participants were asked to collect 100 mL of first morning void (FMV) urine, after which they began collecting urine for the next 24-h, including the first void of the next morning. The 24-h and FMV urine collections were delivered to the study center by courier. The urine was weighed, aliquoted, and stored at −70 °C until being sent to the biorepository where it was also stored at −70 °C.

## Metabolomics analysis

Metabolomic profiling of serum and urine samples was conducted at Metabolon, using ultra-high-performance liquid chromatography (UPLC) coupled with tandem mass spectrometry (MS/MS) to measure a broad range of metabolites. These include endogenous metabolites such as amino acids, carbohydrates, lipids, cofactors, vitamins, and intermediates of energy metabolism, as well as xenobiotics from exogenous sources such as food and drugs. Sample analysis and data processing, including peak alignment and compound identification, have been previously detailed elsewhere [20,32].

Serum and urine samples were analyzed separately. For urine, osmolality measurements were performed by Metabolon using a Fiske 210 Osmometer from 20 μL of each sample. Subsequently, 50 μL of each urine sample was diluted (1:1) with 50 μL of ultrapure $dH_2O$, yielding a final volume of 100 μL. The 100 μL urine samples and all serum samples were prepared using the automated MicroLab STAR system (Hamilton). Recovery standards were added, and proteins were precipitated with methanol, followed by vigorous shaking (Glen Mills GenoGrinder 2000) and centrifugation. The resulting supernatants were dried and reconstituted in recovery solvents containing fixed concentrations of internal standards to ensure consistency across samples, particularly for urine.

Metabolon performed the analyses using a Waters ACQUITY UPLC system coupled with a Thermo Scientific Q-Exactive high-resolution mass spectrometer, equipped with a heated electrospray ionization (ESI) source and an Orbitrap mass analyzer operating at a resolution of 35,000. The extracted metabolites were analyzed using multiple UPLC-MS/MS methods: two reverse-phase UPLC-MS/MS methods in positive ion mode ESI, one reverse-phase UPLC-MS/MS in negative ion mode ESI, and one hydrophilic interaction liquid chromatography (HILIC)-UPLC-MS/MS in negative ion mode ESI. The organic solvent was removed using the TurboVap system (Zymark Corporation). For positive ion mode analysis of both hydrophilic and hydrophobic compounds, samples were gradient-eluted from a C18 column using a mobile phase of water and methanol containing 0.05% perfluoropentanoic acid and 0.1% formic acid. In negative ion mode, a separate C18 column was used, with elution performed using a mixture of methanol, acetonitrile, water, 0.05% perfluoropentanoic acid, and 0.01% formic acid. An additional aliquot underwent analysis in negative ion mode following HILIC column separation (Waters UPLC Bridged Ethylene Hybrid Amide, 2.1 × 150 mm, 1.7 μm), using a gradient of water and acetonitrile containing 10 mM ammonium formate at pH 10.8. The mass spectrometry scan range covered 70–1,000 m/z, with all processes fully automated.

Raw data extraction and metabolite identification were conducted by matching ion chromatographic retention indices, mass-to-charge ratios (m/z), and mass spectral fragmentation patterns against a proprietary reference library of authentic standard spectra. Peak detection and integration were performed using software as previously described [20,33,34]; workflows for setting thresholds for signal-to-noise ratio, peak area, and width were derived from open-source references [33]. To remove batch variability, for each serum and urine metabolite, the raw values were batch-normalized, giving each batch and each metabolite a median of one. Urine measures were divided by their osmolality value and re-scaled to have a median of one. Finally, for each metabolite, the minimum value across all batches in the median scaled data was imputed for the missing values.

## Quality control (QC)

Blinded, replicate, QC serum and urine samples were distributed evenly across batches at a rate of 5%. For serum, coefficients of variation (CVs) were calculated for 1,464 measured metabolites using 72 replicate serum samples from 2 different pooled QC samples ($n$ = 36 QC1; $n$ = 36 QC2). CVs for 1,573 urine metabolites were calculated using 144

replicate urine samples from 2 different pooled QC samples ($n$ = 72 QC1; $n$ = 72 QC2). Additionally, we included 72 duplicate samples from IDATA participants for each sample type to estimate a technical intra-class correlation coefficient (ICC) as well as a six-month, temporal ICC for each metabolite [35]. For a given biospecimen type, the analysis was restricted to metabolites with <90% missingness and <30% CV.

## Statistical analysis

We characterized the distribution of UPF intake by demographic factors. We compared the median dietary intake of macronutrients and micronutrients, based on ASA-24s, and median concentrations of dietary reference biomarkers according to quintiles of the dietary share of UPF using Kruskal–Wallis rank sum test.

For metabolomics analysis, we used averaged measures of UPF intake and metabolites to prioritize longer-term stability over short-term fluctuations in our exposures and to maintain a simple and consistent set of exposure definitions for the correlation and LASSO regression analyses [35]. To account for within-person variability in our analysis, mean metabolite levels per individual, from samples collected 6 months apart, were calculated for a given metabolite and sample type and were then log-transformed using the natural logarithm. We conducted a partial Spearman's rank correlation test to identify serum, 24-h urine, and FMV urine metabolites correlated with percentage of total energy from UPF. This analysis included 952 serum metabolites, and 1,044 and 1,043 from 24-h and FMV urine metabolites, respectively. Partial correlations were adjusted for age (years), sex, race (White, non-Hispanic; African American; Asian; Hispanic), BMI category (18.5 to <25, 25 to <30, 30 to <40, and ≥40 kg/m$^2$), and smoking status (serum cotinine detected: yes, no). To account for multiple testing in the metabolomics analyses, we applied a 1% false discovery rate (FDR) using Benjamini–Hochberg method [36] such that a two-sided $p$-value of <0.01 was considered statistically significant.

To investigate associations between UPF intake and metabolic pathways, we used Fisher's method [37] to combine $p$-values from partial Spearman's rank correlation tests across Metabolon assigned super- and sub-pathways. This method relies on a null distribution created from pseudo replicates, which are derived from a correlation matrix of the metabolites. The null distribution allows us to determine the significance of the combined test statistics. FDR correction was based on the number of pathways considered, and combined two-sided $p$-value threshold of <0.01 was considered statistically significant.

To build a poly-metabolite score predictive of UPF intake (continuous, % energy) using natural log-transformed metabolite data for serum, 24-h urine, or FMV urine, we used Least Absolute Shrinkage and Selection Operator (LASSO) regression [38], applying 10-fold cross-validation to select the optimal $\lambda$ value. To ensure the robustness of our model and prevent overfitting, we divided the dataset into training (80%) and testing (20%) sets. Using the training dataset, we performed LASSO regression to select metabolites. The prediction performance of the selected metabolites was then evaluated using the testing dataset with metrics such as root mean squared error (RMSE) and $R$-squared. Furthermore, we assessed the selection consistency of the metabolites. We applied LASSO on randomly selected 80% subsets of the dataset, repeating this procedure 100 times. The frequency of selection for each metabolite was recorded across the 100 iterations. Finally, we repeated the steps for building a poly-metabolite after dichotomizing the UPF outcome: we coded $Y$ = 1 for subjects with UPF above the third quartile (i.e., >58.2% energy intake from UPF) and $Y$ = 0 (i.e., ≤58.2% energy intake from UPF) for those below. Using this binary outcome, we re-ran the LASSO model and derived predicted scores.

## Poly-metabolite score validation in randomized, controlled, crossover-feeding trial

In a *post-hoc* analysis, we calculated UPF poly-metabolite scores, for EDTA plasma, 24-h urine, and spot urine metabolomics data that was previously generated in a randomized, controlled, crossover-feeding trial (Clinicaltrials.gov NCT03407053; IRB of the National Institute of Diabetes and Digestive and Kidney Diseases approval number 18-DK-0044) [24], as a linear combination of LASSO selected metabolite measures, weighted by corresponding regression coefficients from the LASSO regression models for serum, 24-h urine, and FMV urine in IDATA, respectively. The feeding trial

has been detailed elsewhere. In brief, 20 healthy participants (mean ± standard deviation: age 31 ± 7 years, body mass index [kg/m²] 22 ± 11.6) were domiciled at the NIH Clinical Center and consumed a diet high in UPF (80% energy from UPF) and an unprocessed diet (0% energy from UPF) *ad libitum* for two weeks each. Metabolite levels were analyzed in EDTA plasma, 24-h, and spot urine samples collected at week 2 (i.e., post-intervention) using the same Metabolon UPLC/MS-MS platform [24] that was used to generate metabolomics data in the IDATA Study. We tested for within individual differences in the poly-metabolite score by trial phase (80% and 0% energy from UPF) using two-sided paired *t*-tests [39]. For the poly-metabolite scores derived using the binary UPF outcome, we ran Receiver Operator Characteristic (ROC) curves in training and testing sets to estimate prediction performance using the Area Under the Curve (AUC).

We further considered the poly-metabolite score in a subset of 4 individuals who also participated in a second randomized, crossover-feeding trial, conducted by the same study team, where the participants consumed diets with 30% of energy from UPF, with a varying percentage of calories coming from carbohydrates and fats during each 2-week phase of the 4-week randomized, crossover trial [40]. An ANOVA test was conducted to compare the mean poly-metabolite scores across the three diets. Additionally, two-sided paired *t*-tests were performed to compare the mean poly-metabolite scores between diets with 0% and 30% UPF, as well as between 30% and 80% UPF. The trial metabolomics data is available under the study identifier REQ20250409209843 on MetaboLights (https://www.ebi.ac.uk/metabolights/).

## Results

### Participant characteristics

In our analytic sample, 54% of IDATA participants were between the ages of 60 and 69 years, 51% were female, and 93% self-reported their race/ethnicity as white/non-Hispanic. There was a range of BMI, with 27%, 43% and 30% of individuals classified as normal weight (18.5 to <25 kg/m²), overweight (25 to <30 kg/m²), and obese (>30 kg/m²), respectively. Dietary

**Table 1. Characteristics of IDATA study participants overall and by sex.**

| Characteristic | Overall, N = 718 | Female, N = 365 | Male, N = 353 |
|---|---|---|---|
| Age | | | |
| 50–59 years | 215 (30%) | 138 (38%) | 77 (22%) |
| 60–69 years | 388 (54%) | 184 (50%) | 204 (58%) |
| ≥70 years | 115 (16%) | 43 (12%) | 72 (20%) |
| Race/ethnicity | | | |
| White, non-Hispanic | 667 (93%) | 327 (90%) | 340 (96%) |
| African American | 47 (6%) | 35 (9%) | 12 (3%) |
| Asian | 3 (0.4%) | 3 (0.8%) | 0 (0%) |
| Hispanic | 1 (0.1%) | 0 (0%) | 1 (0.3%) |
| Body mass index | | | |
| 18.5 to <25 kg/m² | 189 (27%) | 126 (35%) | 63 (18%) |
| 25 to <30 kg/m² | 307 (43%) | 126 (35%) | 181 (51%) |
| 30 to <40 kg/m² | 216 (30%) | 112 (31%) | 104 (29%) |
| ≥40 kg/m² | 6 (0.8%) | 4 (1%) | 2 (0.6%) |
| No. ASA-24 completed | | | |
| <4 ASA-24 | 21 (3%) | 16 (4%) | 5 (1%) |
| ≥4 ASA-24 | 697 (97%) | 349 (96%) | 348 (99%) |

Abbreviations: ASA-24s = automated self-administered 24-h dietary assessment tool; IDATA = interactive diet and activity tracking in AARP.

assessment completion rates were high, with 97% of participants completing at least four ASA-24s (Table 1). Only 3 out of 718 participants completed a single ASA-24, while 76% completed all six.

Using repeated ASA-24s, collected over 12 months, the average percentage of daily calories from UPF was 51% in men, 49% in women, 49% in normal weight individuals, and 51% in overweight and obese individuals. We also found that 12-month average macronutrient and micronutrient intake varied substantially with UPF intake. Higher UPF intake was associated with lower average energy intake from protein as well as lower 24-h urinary nitrogen. For example, the average percentage of energy from protein in the highest compared to the lowest quintile of UPF intake was 15% and 18%, respectively. Average daily fiber density varied across quintiles of UPF intake ($p$-value < 0.001), with those in the lowest quintile consuming an average of 7.9 g/1,000 kcal/day and those in the highest consuming an average of 10.7 g/1,000 kcal/day. Similarly, the average dietary intake of several micronutrients, including vitamins A, C, D, and E, as well as iron, zinc, potassium, magnesium, phosphorus, and calcium, varied significantly across quintiles of UPF intake (all $p$-values < 0.05). In contrast, individuals in the highest quintile of UPF intake tended to consume a greater percentage of energy from carbohydrates, added sugars, and saturated fat (all $p$-values < 0.05; Table 2).

### QC results

The median CV for pooled serum QC1 and QC2 were 15.9% (IQR 8.8%–30.6%) and 16.2% (IQR 8.8%–29.8%), respectively. The median CV for pooled urine QC1and QC2 were 18.3% (IQR 12.3%–33.5%) and 18.5% (IQR 13.1%–32.0%), respectively. Median technical ICCs were 0.89, 0.91, and 0.93 for serum, 24-h urine, and FMV urine, respectively. Six-month, temporal ICCs were slightly lower and were more variable across the metabolites; median six-month ICCs were 0.77, 0.64, and 0.50 for serum, 24-h urine, and FMV urine, respectively.

### Metabolite correlations with UPF intake

**Serum.** After FDR correction, mean levels of 191 (155 known and 36 unknown compounds) out of 952 serum metabolites were statistically significantly correlated with UPF intake (S2 Table). The 155 named metabolites included lipids ($n = 56$), amino acids ($n = 33$), carbohydrates ($n = 4$), energy metabolites ($n = 4$), xenobiotics ($n = 33$), cofactors and vitamins ($n = 9$), peptides ($n = 7$), nucleotides ($n = 7$), and partially characterized molecules ($n = 2$). S1 Fig illustrates the correlations between metabolites, with $|r| \geq 0.20$ and FDR-corrected $p$-value < 0.01 for UPF intake, by biospecimen type.

**Urine.** After FDR correction, mean levels of 293 (214 known and 79 unknown) and 237 (170 known and 67 unknown) out of 1,044 and 1,043 metabolites were significantly correlated with UPF intake in 24-h and FMV urine, respectively, with 212 metabolite correlations common to both urine types. The named metabolites in 24-h and FMV urine include lipids ($n = 22$ and 21, respectively; $n$ overlapping = 14), amino acids ($n = 61$ and 50; $n$ overlapping = 44), carbohydrates ($n = 8$ and 7; $n$ overlapping = 7), energy metabolites ($n = 2$ and 1; $n$ overlapping = 1), xenobiotics ($n = 70$ and 51; $n$ overlapping = 48), cofactors and vitamins ($n = 12$ and 12; $n$ overlapping = 12, peptides ($n = 5$ and 5; $n$ overlapping = 4), nucleotides ($n = 10$ and 6; $n$ overlapping = 5), and partially characterized molecules ($n = 23$ and 17; $n$ overlapping = 16) (S3–S4 Tables).

**Serum and urine.** Overall, 470 overlapping metabolites were measured in serum (out of 952 metabolites) and urine (out of 1,044 and 1,043 metabolites). The largest intersection in UPF intake-metabolite correlations ($n = 161$) was for 24-h and FMV urine (Table 3 and S2 Fig). Overall, 49 UPF intake-metabolite correlations were observed in all 3 biospecimen types while 9 were common to serum and 24-h urine, and 3 were common to serum and FMV urine. Notably, 130, 74, and 24 UPF-intake-metabolite correlations were unique to serum, 24-h urine, and FMV urine, respectively.

### Pathway analysis

Across the three biospecimen types, strong and consistent associations with UPF intake were observed for chemical compounds defined by Metabolon as xenobiotics derived from "food component/plant" and related to benzoate metabolism as well as for partially characterized molecules. Additional pathways significantly associated with UPF intake in serum

**Table 2. Indicators of average macronutrient and micronutrient intake[a] and reference biomarkers by quintile of average energy intake from UPF[a] in the IDATA study (N = 718).**

| Indicator of dietary intake | Quintiles of percentage energy from UPF | | | | | Corrected P-value[b] |
| --- | --- | --- | --- | --- | --- | --- |
| | Q1 (11.7%–40.6%) | Q2 (40.7%–47.9%) | Q3 (48.0%–53.3%) | Q4 (53.4%–59.9%) | Q5 (60.0%–82.0%) | |
| | n = 144 | n = 144 | n = 144 | n = 143 | n = 143 | |
| Self-reported measures, median (IQR) | | | | | | |
| Protein (% energy/day) | 17.7 (15.9, 20.4) | 17.5 (15.9, 18.9) | 17.2 (15.4, 19.0) | 15.9 (14.5, 17.6) | 14.6 (12.9, 16.5) | <0.001 |
| Carbohydrates (% energy/day) | 43.6 (38.0, 48.7) | 45.0 (40.2, 49.8) | 46.6 (42.3, 51.5) | 47.4 (43.2, 52.4) | 49.4 (44.7, 54.4) | <0.001 |
| Added sugar (% energy/day) | 6.3 (4.4, 8.3) | 8.4 (6.0, 10.5) | 9.2 (7.3, 12.6) | 10.9 (7.8, 14.1) | 12.9 (9.7, 16.4) | <0.001 |
| Total fat (% energy/day) | 34.9 (30.9, 39.2) | 35.9 (31.3, 39.0) | 35.0 (31.1, 38.6) | 34.7 (30.7, 39.4) | 35.3 (31.5, 38.5) | 0.90 |
| Saturated fat (% energy/day) | 11.1 (9.3, 13.2) | 11.4 (9.8,13.3) | 11.8 (9.5,13.7) | 11.9 (10.1,13.6) | 12.0 (10.6, 13.7) | 0.04 |
| Fiber (g/1,000 kcal/day) | 10.7 (8.5, 13.4) | 9.9 (8.0, 12.1) | 9.3 (7.5, 11.6) | 8.6 (7.0, 10.7) | 7.9 (6.6, 9.8) | <0.001 |
| Vitamin A (µg/1,000 kcal/day) | 417.2 (329.1, 542.8) | 361.5 (301.7, 474.4) | 392.6 (302.4, 493.9) | 386.7 (270.7, 485.1) | 304.0 (247.0, 378.8) | <0.001 |
| Vitamin C (mg/1,000 kcal/day) | 61.5 (43.7, 82.1) | 47.8 (35.0, 66.7) | 42.0 (28.2, 62.7) | 38.8 (26.1, 53.8) | 33.8 (21.9, 49.2) | <0.001 |
| Vitamin D (µg/1,000 kcal/day) | 2.5 (1.6, 3.9) | 2.2 (1.5, 3.3) | 2.0 (1.6, 3.0) | 2.0 (1.3, 2.9) | 1.7 (1.2, 2.4) | <0.001 |
| Vitamin E (mg/1,000 kcal/day) | 4.6 (3.6, 6.4) | 4.1 (3.4, 5.3) | 4.0 (3.2, 4.7) | 3.8 (3.2, 4.6) | 3.4 (3.0, 4.3) | <0.001 |
| Iron (mg/1,000 kcal/day) | 7.6 (6.5, 8.7) | 7.7 (6.6, 9.1) | 7.5 (6.5, 8.5) | 7.9 (6.5, 9.5) | 7.3 (6.2, 8.6) | 0.04 |
| Zinc (mg/1,000 kcal/day) | 6.2 (5.5, 7.6) | 6.4 (5.4, 7.5) | 6.2 (5.4, 7.1) | 5.9 (5.1, 7.2) | 5.4 (4.6, 6.2) | <0.001 |
| Potassium (mg/1,000 kcal/day) | 1678.0 (1485.5, 1854.8) | 1524.3 (1339.9, 1703.6) | 1444.6 (1312.7, 1656.3) | 1397.3 (1199.5, 1541.4) | 1225.4 (1072.3, 1380.3) | <0.001 |
| Phosphorus (mg/1,000 kcal/day) | 724.0 (653.1, 807.9) | 708.2 (645.3, 779.2) | 695.7 (640.2, 765.5) | 665.1 (583.6, 716.7) | 608.4 (555.2, 675.3) | <0.001 |
| Magnesium (mg/1,000 kcal) | 180.5 (157.2, 215.5) | 168.3 (144.4, 196.5) | 159.5 (142.2, 180.9) | 148.2 (129.9, 171.1) | 132.2 (118.9, 152.1) | <0.001 |
| Calcium (mg/1,000 kcal/day) | 480.8 (388.9, 581.8) | 465.8 (391.5, 569.5) | 471.9 (383.5, 545.4) | 445.4 (370.2, 573.2) | 415.0 (336.6, 497.1) | <0.001 |
| Reference biomarker measures, median (IQR) | | | | | | |
| Urinary sodium (mg/day) | 3624.5 (2572.9, 5138.6) | 4046.7 (3092.4, 5556.3) | 3938.4 (2882.9, 5522.3) | 4002.9 (2723.1, 5296.5) | 3801.7 (2958.0, 5107.5) | 0.09 |
| Urinary potassium (mg/day) | 3384.1 (2719.9, 4477.4) | 3584.2 (2685.1, 4502.5) | 3334.8 (2470.5, 4351.2) | 3052.1 (2232.8, 3844.9) | 2762.2 (2030.5, 3593.5) | <0.001 |
| Urinary nitrogen (g/day) | 90.5 (69.8, 109.0) | 96.0 (72.9, 122.9) | 91.0 (67.7, 112.5) | 86.8 (68.6, 109.4) | 77.8 (62.7, 103.6) | 0.01 |

[a]Average intake estimated from up to 6 ASA-24s collected over a 12-month period.

[b]Kruskal–Wallis rank sum test with multiple testing corrected for using the Benjamini–Hochberg method.

Abbreviations: ASA-24s = automated self-administered 24-h dietary assessment tool; IDATA = interactive diet and activity tracking in AARP; Q = quintile; UPF = ultra-processed food.

**Table 3. Interssection of metabolites correlated with percentage energy from UPF (corrected P-value of <0.01) by biospecimen type and chemical class in the IDATA Study (n = 718).**

| Super pathway | Biospecimen type | | | | | | | Total |
| --- | --- | --- | --- | --- | --- | --- | --- | --- |
| | 24-h + FMV urine | Serum only | 24-h urine only | Serum + 24-h + FMV urine | FMV urine only | Serum + 24-h urine | Serum + FMV urine | |
| Amino acid | 34 | 21 | 16 | 10 | 5 | 1 | 1 | 88 |
| Carbohydrate | 5 | 2 | 1 | 2 | – | – | – | 10 |
| Cofactors and vitamins | 8 | 5 | – | 4 | – | – | – | 17 |
| Energy | – | 3 | 1 | 1 | – | – | – | 5 |
| Lipid | 10 | 49 | 7 | 4 | 5 | 1 | 2 | 78 |
| Nucleotide | 3 | 4 | 4 | 2 | 1 | 1 | – | 15 |
| Partially characterized | 15 | 1 | 7 | 1 | 1 | – | – | 25 |
| Peptide | 3 | 6 | 2 | 1 | 1 | – | – | 13 |
| Unknown | 48 | 22 | 17 | 11 | 8 | 3 | – | 109 |
| Xenobiotics | 35 | 17 | 19 | 13 | 3 | 3 | – | 90 |
| **Total** | **161** | **130** | **74** | **49** | **24** | **9** | **3** | **450** |

and urine samples involved: amino acid metabolism (i.e., methionine, cysteine, SAM, and taurine metabolism; tyrosine metabolism; urea cycle -arginine and proline metabolism; tryptophan metabolism; glutathione metabolism; lysine metabolism; alanine and aspartate metabolism; and leucine, isoleucine, and valine metabolism), lipid metabolism (i.e., fatty acid, dicarboxylate; fatty acid, dihydroxy; phospholipid metabolism; and fatty acid metabolism (acyl carnitine, dicarboxylate)), carbohydrate metabolism (i.e., pentose metabolism; glycolysis, gluconeogenesis, and pyruvate metabolism), cofactor and vitamin metabolism (i.e., tocopherol metabolism; and ascorbate and aldarate metabolism), and nucleotide metabolism (i.e., uracil containing pyrimidine metabolism; and purine metabolism containing (hypo)xanthine/inosine). Energy metabolism pathways, specifically the TCA cycle, showed moderate but consistent associations across serum and urine, pointing to possible alterations in energy production processes with high UPF intake (S5 Table).

### Poly-metabolite score generation in IDATA Study

For poly-metabolite scores of continuous UPF intake, we selected 28, 33, and 23 metabolites for serum, 24-h urine, and FMV urine, respectively, using LASSO regression (Fig 2 and S6 Table). RMSE and Mean Absolute Error (MAE) values were similar in testing and training datasets, for each biospecimen type, suggesting that the prediction models were not overfit and performing well; additionally, correlations between the scores and UPF intake were moderate to strong ($r \geq 0.47$) [19] in testing and training datasets for each biospecimen type (S7 Table). In serum and 24-h urine analyses, three common metabolites were selected by LASSO regression ≥60 times out of 100 iterations, making them robust predictors [41] of UPF intake (Fig 3). These include two amino acids (i.e., (S)C(S)S-S-Methylcysteine sulfoxide [$r_s$ = −0.19 to −0.23] and N2,N5-diacetylornithine [$r_s$ = −0.26 to −0.27]); one partially characterized molecule (i.e., pentoic acid [$r_s$ = −0.28 to −0.32), and one carbohydrate (i.e., N6-carboxymethyllysine [$r_s$ = 0.15 to 0.21]). For poly-metabolite scores of binary UPF intake, we selected 5, 12, and 9 metabolites for serum, 24-h urine, and FMV urine, respectively (S6 Table). The AUCs for training and test sets were 0.75 and 0.66 for serum, 0.78 and 0.72 for 24-h urine, and 0.77 and 0.68 for FMV urine, indicating that poly-metabolite scores offer moderate discrimination between those in the highest quartile and those in the lower three quartiles of intake based on the IDATA population distribution.

### Poly-metabolite score validation in randomized, controlled, crossover-feeding trial

We applied the serum and urine poly-metabolite scores, developed in IDATA, *post-hoc* to existing plasma and urine metabolomics data generated in an independent, randomized, controlled, crossover-feeding trial. For the score calculation, 26 out of 28 IDATA-selected serum metabolites were measured in the trial in EDTA plasma as were all selected urine metabolites. We found that the blood- and urine-based poly-metabolite scores differed substantially, within individual, between the 80% and 0% energy from UPF diet phases (all *p*-value for paired *t* test < 0.001; Fig 4). In contrast, some but not all of the individual metabolites included in the poly-metabolite scores differed, within individual, between diet phases, (S8 Table) suggesting that the score may be better suited as a candidate biomarker of UPF intake than any one individual metabolite. In the subset of four individuals who participated in a second feeding trial of diets with 30% energy from UPF, we found that the poly-metabolite score for 24-h urine increased stepwise, within individual, with increasing energy from UPF; for EDTA plasma and spot urine, ploy-metabolite scores were similar for diets low (i.e., 0 and 30% energy) in UPF and significantly increased for diets high in (80% energy) UPF (S3–S5 Figs).

### Discussion

Understanding how metabolites are associated with UPF intake and identifying poly-metabolite scores predictive of UPF intake in free-living individuals with varying diets has the potential to complement, extend, and advance research on UPF intake and human health. In the IDATA Study, we identified metabolites significantly correlated with UPF intake using averaged, repeated measures of metabolites and dietary intake, assessed via multiple ASA-24s, in 718 generally healthy adults. Moreover, we developed blood and urine poly-metabolite scores for UPF intake, using untargeted serum, 24-h

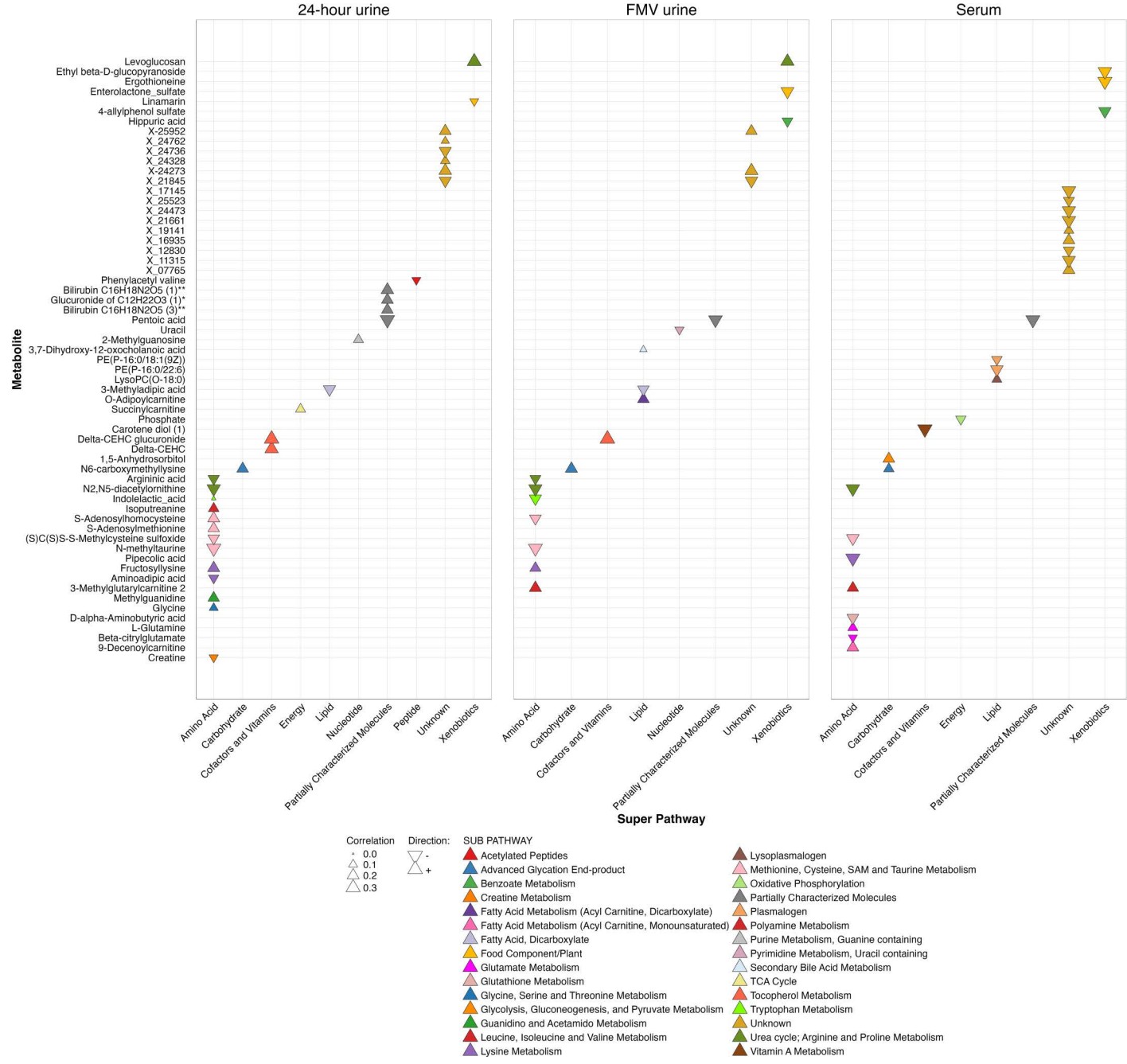

**Fig 2. Relationship between UPF intake and LASSO-selected metabolites by biospecimen type and metabolic pathway.** The plot displays the correlations between individual metabolites and UPF intake in 24-h urine, FMV urine, and serum samples in the IDATA Study (*n* = 718). Metabolites are categorized by chemical class or super pathway, shown on the *x*-axis, while individual metabolites are listed on the *y*-axis. The direction and magnitude of correlations are represented by triangles, where upward triangles indicate positive correlations, and downward triangles indicate negative correlations. The size of each triangle corresponds to the correlation coefficient, and colors indicate the Metabolon assigned metabolic sub-pathway. Only metabolites selected by LASSO regression for at least one biospecimen type are displayed. Abbreviations: FMV = first morning void; IDATA = interactive diet and activity tracking in AARP; LASSO = least absolute shrinkage and selection operator; UPF = ultra-processed food.

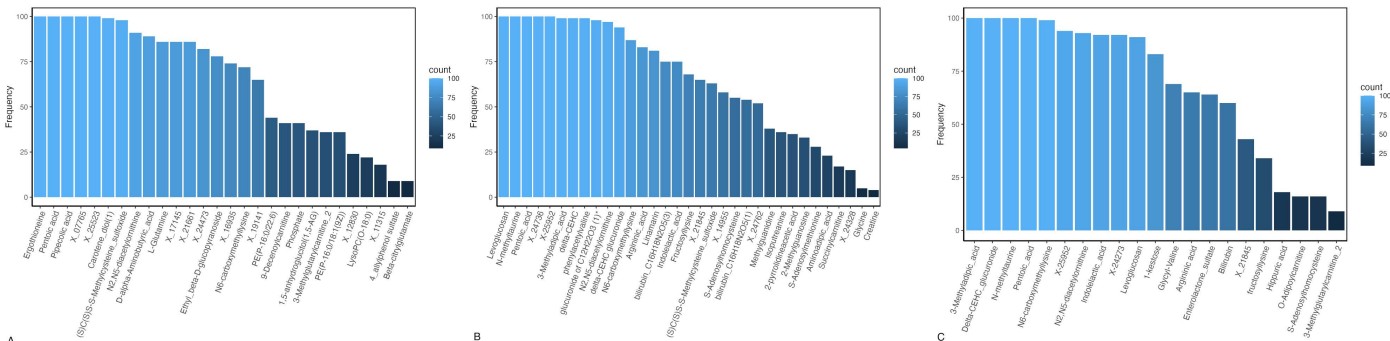

**Fig 3. Metabolites selected using LASSO regression as predictive of 12-month average UPF intake (% energy) by biospecimen type.** This plot shows the number of times a metabolite was selected by the LASSO regression model as predictive of UPF intake (% energy) in **(A)** serum, **(B)** 24-h urine, and (C) FMV urine in the IDATA Study (*n* = 718). For each biospecimen type, we applied LASSO to randomly selected 80% subsets of the dataset, repeating this procedure 100 times. The *y*-axis represents the number of times each metabolite was selected out of 100 iterations, and the *x*-axis displays the names of the LASSO selected metabolites. Abbreviations: FMV = first morning void; IDATA = interactive diet and activity tracking in AARP; LASSO = least absolute shrinkage and selection operator; UPF = ultra-processed food.

urine, and FMV urine metabolomics data and a rigorous statistical approach, including training and testing datasets, to ensure the robustness of our results. We then applied these scores to EDTA plasma, 24-h urine, and spot urine metabolomics data, respectively, that had been previously generated in an independent, randomized, controlled, crossover, feeding trial. We demonstrated the validity of these poly-metabolite scores by showing that they differed, within individual, between consumption of diets high in (80% energy) versus to void of UPF (0% energy).

Overall, we identified hundreds of metabolites in serum (*n* = 191) and urine (*n* = 293 in 24-h urine and *n* = 237 in FMV) that were significantly correlated with UPF intake. These UPF-correlated metabolites are involved in numerous and diverse biological pathways, underscoring the complex impact of diet on the metabolome. For instance, among the metabolites that were positively correlated with UPF intake, there were acylcarnitine derivatives, such as 9-decenoylcarnitine, and sphingolipids, such as the ceramides d18:1/18:0 and d18:1/20:0. Higher serum levels of acylcarnitine have been found in people who consume meat compared to vegetarians [22,42,43]. In contrast, lower levels of the ceramides, d18:1/18:0 and d18:1/20:0, have been associated with better adherence to healthy dietary patterns [44]. An even larger number of metabolites were negatively correlated with UPF intake, including β-cryptoxanthin, which has been identified as a candidate biomarker of fruit and vegetable consumption [22,45].

Using LASSO regression models, we identified a set of circulating and urinary metabolites that may serve collectively as predictive biomarkers for UPF intake. In total, we selected 28 serum, 33 24-h urine, and 23 FMV urine metabolites as predictors of UPF intake. Of these metabolites, three were robust (selected ≥60 times out of 100 iterations) biomarkers and were common to both serum and 24-h urine including two amino acids (i.e., (S)C(S)S-S-Methylcysteine sulfoxide and N2,N5-diacetylornithine) which were both negatively correlated with UPF intake, and a carbohydrate (i.e., N6-carboxymethyllysine), which was positively correlated with UPF intake. (S)C(S)S-S-Methylcysteine sulfoxide has been identified as a biomarker of cruciferous vegetable intake [46] indicating that dietary patterns high in UPF intake tend to be low in cruciferous vegetable intake. Serum levels of N6-carboxymethyllysine, an advanced glycation end-product (i.e., a family of compounds that form when sugars react with proteins, lipids or nucleic acids), have been associated with diabetes incidence and severity as well as with risk of other cardiometabolic diseases [47–49]. The positive correlation between N6-carboxymethyllysine and UPF intake in our study suggests a potential link between a dietary pattern high in UPF and type 2 diabetes risk [50] through this metabolite or related metabolic pathways. This aligns with a systematic umbrella review of existing meta-analyses, which reported that greater exposure to UPF is associated with a higher risk of some adverse health outcomes, including type 2 diabetes [12]. Moreover, this suggests the utility in using nutritional

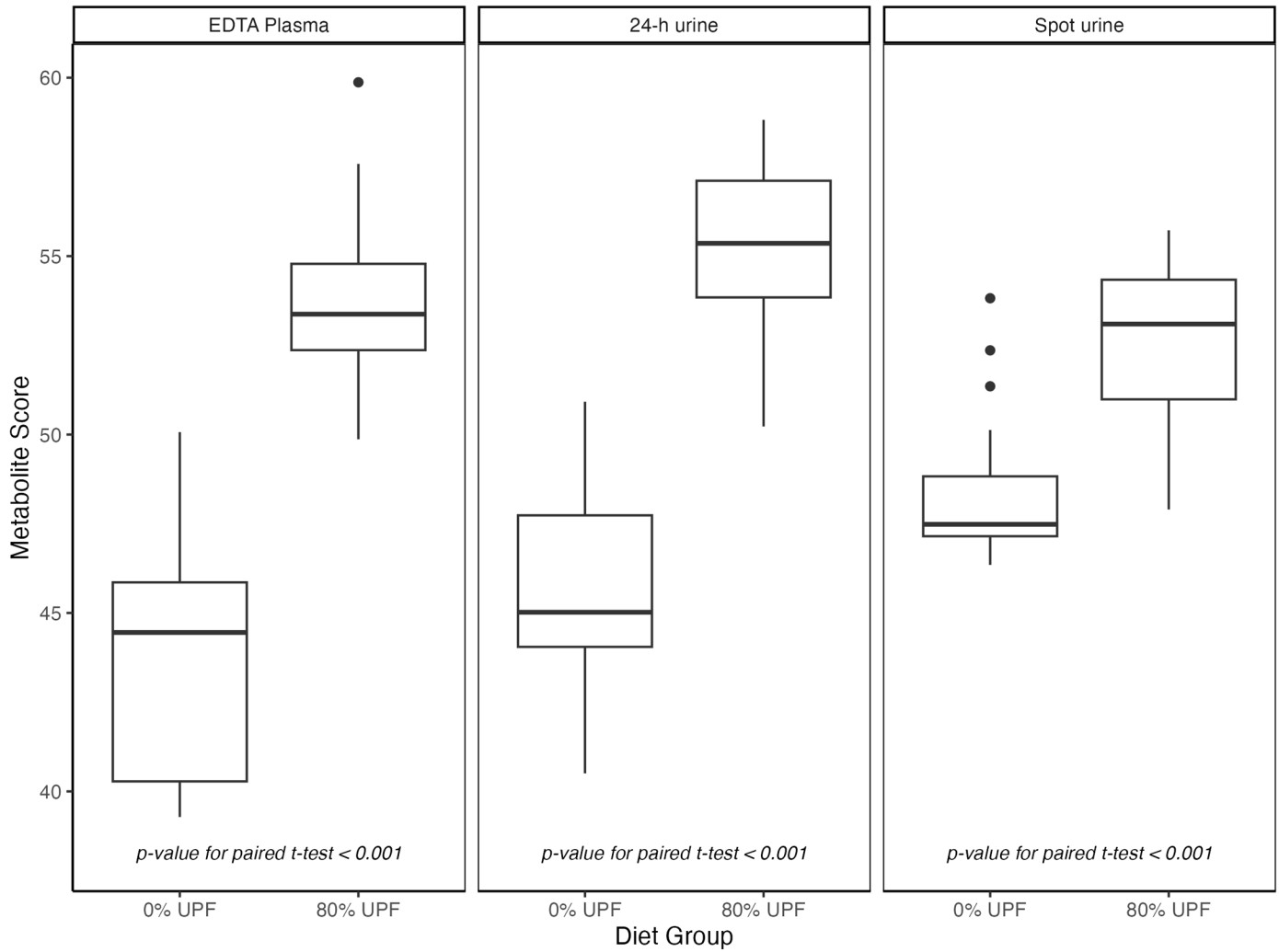

**Fig 4. Post-hoc comparison of UPF poly-metabolite scores by diet phase in a randomized, controlled, crossover-feeding trial of domiciled adults ($N$ = 20).** Box plots show the distribution of poly-metabolite scores, developed in the IDATA study, generated using EDTA plasma and urine metabolomics data from an independent, randomized, controlled, crossover-feeding trial of 20 domiciled healthy participants, who consumed *ad libitum*, for 2 weeks each, an ultra-processed food dietary pattern (i.e., with 80% energy from UPF) and an unprocessed dietary pattern (i.e., 0% energy from UPF). The box represents the interquartile range, the center line represents the median, the whiskers show the minimum and maximum values within 1.5 times the interquartile range, and data points outside the whiskers show poly-metabolite score outliers. Differences in poly-metabolite scores, within individual, between the two diet phases of the trial were tested using paired *t*-tests. Abbreviations: FMV urine = first morning void urine, UN-DP, unprocessed food dietary pattern, UPF-DP, ultra-processed food dietary pattern.

metabolomics not only to improve exposure measures but also to provide novel insight into biological mechanisms underlying potential UPF-disease associations.

Our findings suggest that the metabolomic profiles we identified reflect a dietary pattern characterized not only by high UPF consumption but also by low intake of whole foods, including fresh fruits and vegetables. In the serum set, three metabolites (out of six) in the chemical class, cofactors and vitamins, were negatively correlated with UPF intake; β-cryptoxanthin and carotene diol [1] and [2] are provitamin A carotenoids found in citrus fruits and some vegetables, such as red peppers, that have been identified as candidate biomarkers of fruit and vegetable intake [22,45]. β-cryptoxanthin has also been associated with adherence to a healthy dietary pattern [22,45,51]. In line with these metabolite findings, in

IDATA, we observed that average daily dietary intake of key vitamins and minerals was lower among those with higher UPF intake. This observation aligns with prior research indicating an inverse association between UPF intake and diet quality among both children and adults [52].

The urine metabolites, selected by LASSO regression models, similarly included a diverse array of compounds. Notably, the exogenously derived metabolite, levoglucosan, which was positively correlated with UPF intake in our study, is a combustion breakdown product of cellulose, which has been used as a biopolymer in food packaging [53], and is excreted unmetabolized in urine after exposure [54]. This finding suggests that higher intake of UPF may expose individuals to food contact substances, including some from packaging materials. It is, however, important to note that this dimension of UPF exposure was reflected in urinary and not circulating metabolites. This is an important consideration for future researchers and highlights the need for dietary biomarker discovery not only in blood but also in urine and stool.

The urine metabolites that were selected by LASSO regression were similar to those selected for 24-h urine but more limited in number. Overall, 212 UPF-metabolite correlations, which were generally similar in both magnitude and direction, were observed in both urine types demonstrating that FMV metabolite-UPF correlations are reflective of those for 24-h urine. This is important since 24-h collections are not feasible in most large epidemiological studies. We also show that while there is some overlap between blood and urine for UPF-metabolite correlates and related scores, most are unique to either blood or urine. Distinct metabolites and metabolomic profiles by biospecimen type may also translate to differences in mechanistic insight into potential UPF-health associations in etiologic studies. Our IDATA findings are consistent with a recent short-term randomized, controlled, crossover, domiciled feeding trial [24], which found that hundreds of circulating (257 out of 993) and urine metabolites (606 out of 1,279) differed within individuals between dietary patterns containing 80% UPF and 0% UPF in post-intervention samples. We found that a majority of the metabolites identified as potential candidate biomarker in the feeding trial (i.e., 195 out of 257 circulating metabolites and 334 out of 606 urine metabolites) were also detected in IDATA samples. Of the 59 serum and 126 urine metabolites that were statistically significant in the IDATA, 85% and 79% were associated with UPF intake in the same direction (S9-S10 Tables). Three metabolites (i.e., N6-carboxymethyllysine, (S)C(S)S-S-Methylcysteine sulfoxide, and pentoic acid) that were selected as robust predictors of UPF intake in both serum and urine were among those reported in the feeding trial. When we tested our IDATA-derived poly-metabolite scores for UPF intake in this same feeding trial, we found that they differed between dietary patterns containing 80% UPF versus 0% UPF, within individual, indicating that our novel poly-metabolite scores could be used as an objective and complementary measure of UPF intake in future epidemiological investigations.

Our study has notable strengths, which contribute to its scientific rigor. Firstly, diet over 12 months was assessed via multiple validated 24-h dietary recalls, with more than 97% of participants completing four or more. This helped minimize measurement error due to random within-person day-to-day variation in dietary intake, and it also allowed for more accurate and detailed dietary data collection than a food frequency questionnaire. While alternative statistical methods could account for within-subject variability differently, our primary goal was biomarker selection rather than modeling longitudinal dietary intake trends. To our knowledge, there are no readily available statistical tools that integrate variable selection with repeatedly measured exposures and outcomes in a high-dimensional metabolomics setting. Given this limitation, we adopted an averaging approach, which provides a practical and interpretable solution for identifying robust biomarker signatures associated with average dietary intake. Secondly, we measured more than a thousand metabolites, related to wide range of metabolic pathways, in serial serum and urine samples collected from the same individuals 6-months apart. Serial metabolite measurements again helped to minimize measurement error owing to within-person variability in metabolite measures over time. Thirdly, we not only successfully replicated many of the associations between UPF and metabolites identified in a prior feeding trial but also demonstrated that our poly-metabolite scores effectively differentiate, within individuals, between diets high in UPF and void of UPF in that same trial. While our IDATA Study population lacked diversity, the participants consumed a range of diets, and diets high in UPF were not the same, suggesting our findings may be generalizable to other populations.

Our study is not without limitations. For example, in the IDATA Study dietary recalls could not be reliably matched with specific biospecimen measurements as recalls were, by design, unannounced. Future studies, including feeding trials, with precise pairing of dietary intake and biospecimen collection are needed to better capture temporal dynamics. Despite the small sample size of the randomized, controlled, cross-over feeding trial, which may have contributed to the observed variation in our results, we internally validated the poly-metabolite score within the IDATA Study using LASSO regression with repeated training and testing splits. This approach ensured that our biomarker selection was not overly reliant on a single dataset and remained robust across different subsamples. While our findings lay an important foundation for exploring the potential of multi-metabolite profiles as objective markers of UPF intake, they will likely need to be iteratively improved with expanding datasets similar to polygenic risk scores. Furthermore, there is no universally accepted threshold for defining a diet high in UPF; thus, our ROC Curve analysis, which demonstrated moderate discrimination between those above and below the 75th percentile of intake, is population specific and may not be directly comparable to studies using alternative definitions or to populations with varying distributions of UPF intake.

In summary, we show that dietary patterns with higher UPF intake are associated with distinctive metabolomic profiles in both serum and urine. Future studies are needed to validate our UPF poly-metabolite scores in more diverse populations with varying diets and to explore whether integrating poly-metabolite scores with self-reported dietary data can reduce measurement error and improve risk prediction. Finally, research on the prospective association between UPF-correlated metabolites, poly-metabolite scores, and disease risk is warranted.

## Supporting information

**S1 Fig. Heatmap of metabolites correlated with UPF intake in (A) serum, (B) 24-h, and (C) FMV urine.** Metabolite-UPF intake (% energy) correlations were estimated using Spearman partial correlation, adjusted for age (continuous), sex (men, women), smoking (cotinine detected: yes, no), race (White, non-Hispanic, African American, Asian, Hispanic) and BMI (18.5 to <25, 25 to <30, 30 to <40, ≥40 kg/m$^2$) in the IDATA Study ($N$ = 718). Multiple testing was corrected for using the Benjamini–Hochberg method; statistical significance was defined as a corrected $p$-value of <0.01. Correlations between metabolites with $|r| \geq 0.20$ and FDR-corrected $p$-value <0.01 are visualized in the heatmap. The dark purple color shows strong negative metabolite-metabolite correlations and the dark red color shows strong positive metabolite-metabolite correlations. The dendrogram uses hierarchical clustering and shows that metabolites generally cluster according to the direction of their correlation with UPF intake. Abbreviations: UPF = ultra-processed food.
(TIFF)

**S2 Fig. Intersection of metabolites correlated with UPF intake by biospecimen type and chemical class.** The UpSet plot shows the intersections of metabolites associated with UPF intake (% energy) after multiple testing was corrected for using the Benjamini–Hochberg method; statistical significance was defined as a corrected $p$-value of <0.01. Vertical bars represent the count of shared metabolites across these biospecimen types, with horizontal bars reflecting the total count of metabolites associated with UPF intake in each biospecimen type. The matrix at the bottom indicates intersection by biospecimen type. Abbreviations: UPF = ultra-processed food.
(TIFF)

**S3 Fig. Comparison of EDTA plasma UPF poly-metabolite scores by diet phase for participants ($n$=4) in two distinct randomized, controlled, crossover, feeding trials, of diets that were 0%, 30%, and 80% energy from UPF.** Box plots show the distribution of poly-metabolite scores, developed in the IDATA study using serum, and generated using EDTA plasma from domiciled healthy participants ($n$ = 4) who completed two independent, randomized, controlled, crossover-feeding trials. In the first trial, participants consumed *ad libitum*, for 2 weeks each, an UPF-DP (i.e., with 80% energy from UPF) and an UN-DP pattern (i.e., 0% energy from UPF). In the second trial, participants, consumed *ad libitum* a plant-based low-fat and an animal-based ketogenic dietary pattern that were low-carbohydrate/low-fat and

contained 30% energy from UPF. The box represents the interquartile range, the center line represents the median, the whiskers show the minimum and maximum values within 1.5 times the interquartile range, and data points outside the whiskers show poly-metabolite score outliers. An ANOVA test ($P$-value < 0.001) was conducted to compare the mean poly-metabolite scores across the three diet phases with paired samples, and paired $t$-tests were performed to compare the mean poly-metabolite scores between 0% and 30% UPF ($P$-value = 0.60) as well as between 30% and 80% UPF ($P$-value < 0.01). Abbreviations: UN-DP = unprocessed food dietary pattern, UPF-DP = ultra-processed food dietary pattern.
(TIFF)

**S4 Fig. Comparison of 24-h urine UPF poly-metabolite scores by diet phase for participants ($n$=4) in two distinct randomized, controlled, crossover, feeding trials, of diets that were 0%, 30%, and 80% energy from UPF.** Box plots show the distribution of poly-metabolite scores, developed in the IDATA study using 24-h urine, and generated using 24-h urine from domiciled healthy participants ($n$ = 4) who completed two independent, randomized, controlled, crossover-feeding trials. In the first trial, participants consumed *ad libitum*, for 2 weeks each, an UPF-DP (i.e., with 80% energy from UPF) and an UN-DP (i.e., 0% energy from UPF). In the second trial, participants, consumed *ad libitum* a plant-based low-fat and an animal-based ketogenic dietary pattern that were low-carbohydrate/low-fat and contained 30% energy from UPF. The box represents the interquartile range, the center line represents the median, the whiskers show the minimum and maximum values within 1.5 times the interquartile range, and data points outside the whiskers show poly-metabolite score outliers. An ANOVA test ($P$-value = 0.001) was conducted to compare the mean poly-metabolite scores across the three DPs and paired $t$-tests were performed to compare the mean poly-metabolite scores between 0% and 30% UPF ($P$-value = 0.06) as well as between 30% and 80% UPF ($P$-value = 0.04). Abbreviations: UN-DP = unprocessed food dietary pattern, UPF-DP = ultra-processed food dietary pattern.
(TIFF)

**S5 Fig. Comparison of spot urine UPF poly-metabolite scores by diet phase for participants ($n$=4) in two distinct randomized, controlled, crossover, feeding trials, of diets that were 0%, 30%, and 80% energy from UPF.** Box plots show the distribution of poly-metabolite scores, developed in the IDATA Study using FMV urine, and generated using spot urine from domiciled healthy participants ($n$ = 4) who completed two independent, randomized, controlled, crossover-feeding trials of. In the first trial, participants consumed *ad libitum*, for 2 weeks each, an UPF-DP (i.e., with 80% energy from UPF) and an UN-DP (i.e., 0% energy from UPF). In the second trial, participants, consumed *ad libitum* a plant-based low-fat and an animal-based ketogenic dietary pattern that were low-carbohydrate/low-fat and contained 30% energy from UPF. The box represents the interquartile range, the center line represents the median, the whiskers show the minimum and maximum values within 1.5 times the interquartile range, and data points outside the whiskers show poly-metabolite score outliers. An ANOVA test ($P$-value < 0.01) was conducted to compare the mean poly-metabolite scores across the three DPs and paired $t$-tests were performed to compare the mean poly-metabolite scores between 0% and 30% UPF ($P$-value = 0.90) as well as between 30% and 80% UPF ($P$-value = 0.02). Abbreviations: FMV urine = first morning void urine, UN-DP = unprocessed food dietary pattern, UPF-DP = ultra-processed food dietary pattern.
(TIFF)

**S1 Table. Contribution of the top 10 food codes to percentage of energy from ultra-processed foods (NOVA 4).**
(XLSX)

**S2 Table. Serum metabolites significantly correlated with percentage energy from ultra-processed food intake.**
(XLSX)

**S3 Table. 24-h urine metabolites significantly correlated with percentage energy from UPF intake.**
(XLSX)

**S4 Table. First morning void urine metabolites significantly correlated with percentage energy from ultra-processed food intake.**
(XLSX)

**S5 Table. Pathway analysis using Fisher's test method to combine *p*-values for the correlation between metabolites and percentage energy from ultra-processed food intake by Metabolon defined super- and sub-pathways.**
(XLSX)

**S6 Table. Calculated coefficients for metabolites selected by LASSO regression model for continuous UPF intake (% energy) and dichotomized (<75th or ≥75th percentile) UPF intake.**
(XLSX)

**S7 Table. LASSO regression analysis metrics for continuous UPF intake (% energy) by biospecimen type.**
(XLSX)

**S8 Table. Paired *t* test results for LASSO selected metabolites (continuous UPF intake) and diet phase (80% UPF versus 0% UPF) in a randomized, controlled, crossover-feeding trial by biospecimen type.**
(XLSX)

**S9 Table. Overlap in EDTA plasma and serum metabolites in the randomized, controlled, crossover-feeding trial and the IDATA study, respectively.**
(XLSX)

**S10 Table. Overlap in urine metabolites in the randomized, controlled, crossover-feeding trial and the IDATA study.**
(XLSX)

**S1 Checklist. Consort checklist.**
(DOC)

## Acknowledgments

The authors thank the National Cancer Institute Interactive Diet and Activity Tracking in AARP (IDATA) Study team. This project was approved under proposal IDATAB-2020-1014. The findings in this article are those of the authors and do not necessarily represent the official position of the National Institutes of Health or USDA.

## Author contributions

**Conceptualization:** Leila Abar, Erikka Loftfield.

**Data curation:** Leila Abar, Sang Kyu Lee, Erikka Loftfield.

**Formal analysis:** Leila Abar, Sang Kyu Lee, Caitlin P. O'Connell, Erikka Loftfield.

**Funding acquisition:** Erikka Loftfield.

**Investigation:** Leila Abar, Erikka Loftfield.

**Methodology:** Leila Abar, Hyokyoung G. Hong, Erikka Loftfield.

**Project administration:** Erikka Loftfield.

**Resources:** Lisa Kahle, Erikka Loftfield.

**Supervision:** Erikka Loftfield.

**Validation:** Erikka Loftfield.

**Visualization:** Leila Abar, Sang Kyu Lee, Caitlin P. O'Connell, Erikka Loftfield.

**Writing – original draft:** Leila Abar.

**Writing – review & editing:** Leila Abar, Eurídice Martínez Steele, Steven C. Moore, Eleanor Watts, Caitlin P. O'Connell, Charles E. Matthews, Kirsten A. Herrick, Kevin D. Hall, Lauren E. O'Connor, Neal D. Freedman, Rashmi Sinha, Erikka Loftfield.

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
