## [Editor Report · Decision Letter 0]

19 Aug 2024

Dear Dr Abar, 

Thank you for submitting your manuscript entitled "Serum and urine metabolomic profiles of long-term ultra-processed food intake in the longitudinal Interactive Diet and Activity Tracking in AARP Study" for consideration by PLOS Medicine.

Your manuscript has now been evaluated by the PLOS Medicine editorial staff and I am writing to let you know that we would like to send your submission out for external peer review.

Please re-submit your manuscript within two working days, i.e. by Aug 21 2024.

Feel free to email me at atosun@plos.org or us at plosmedicine@plos.org if you have any queries relating to your submission.

Kind regards,

Alexandra Tosun, PhD

Associate Editor

PLOS Medicine

---

## [Decision Letter · Decision Letter 1]

22 Jan 2025

Dear Dr Abar,

Many thanks for submitting your manuscript "Serum and urine metabolomic profiles of long-term ultra-processed food intake in the longitudinal Interactive Diet and Activity Tracking in AARP Study" (PMEDICINE-D-24-02724R1) to PLOS Medicine. The paper has been reviewed by subject experts and a statistician; their comments are included below and can also be accessed here: [LINK]

As you will see, the reviewers raise specific concerns including around the statistical reporting and the potential broader applicability of the findings in your study. After discussing the paper with the editorial team and an academic editor with relevant expertise, I'm pleased to invite you to revise the paper in response to the reviewers' comments. We plan to send the revised paper to some or all of the original reviewers, and we cannot provide any guarantees at this stage regarding publication.

We ask that you submit your revision by Feb 12 2025 11:59PM. However, if this deadline is not feasible, please contact me by email, and we can discuss a suitable alternative.

Don't hesitate to contact me directly with any questions (rkirk@plos.org). 

Best regards, 

Rebecca Kirk

On behalf of:

Louise Gaynor-Brook, MBBS PhD 

Senior Editor

PLOS Medicine

lgaynor@plos.org

Comments from the academic editor:

The most important strength of this study is that the authors identified metabolite predictors of UPF and tested its utility in the cross-over RCT, combining their observational work with the independent RCT data. However, the RCT requires clarification and improved statistical reporting.

UPFs are known to be misclassified. Healthy items, such as mass-produced wholegrain snacks, can be UPF. The authors should analyze and document what food items were predominant in their analyses.

Comments from the reviewers: 

Reviewer #1: This research identified serum and urine biomarkers that are associated with consumption of ultra processed foods (UPF), as determined through correlation-based analyses of self-reported dietary data. Identified biomarkers were used to generate a score, that was validated against two small crossover trials with known levels of UPF consumption. 

Overall, this manuscript is clear, cogent, and well presented. The authors have argued the importance of the research convincingly and communicated complex ideas around biomarkers and metabolomics well. There are some opportunities to more clearly define and justify their decisions pertaining to statistical analyses and in some cases important modelling details are missing. They may also consider employing more sophisticated or contemporary statistical approaches. In particular, their decision to analyse longitudinal data by averaging first then simply analysing the averages has prompted a number of questions and may be worth revisiting. 

Detailed feedback on the manuscript is provided below. My primary focus has been methodological, but I have made a few broader points as well. All items are major unless indicated otherwise. 

1. General 

1.1 Consider making clear in both the title and abstract that there are tight constraints on the population of participants being researched, notably age. Not referring to this potentially misleads the reader into understanding this to be about, say, all adults. 

1.2 [MINOR] Given its salience to the paper, consider tweaking the title to incorporate the secondary as well as the primary aim. 

2. Abstract

2.1 Please revise based on the provided feedback. 

3. Introduction

3.1 [MINOR] "The integration of dietary data, measured via self-report using validated assessment tools, with biomarker measures can control confounding, minimize measurement errors, and enhance statistical power in analyses of diet and disease" I understand this is just background, but is it correct? The methods described in the cited papers appear to adjust for measurement error (not "minimise" it) and I'm struggling with the claim that integration of self-report dietary with biomarker data can "control confounding". 

3.2 "Currently, nutritional metabolomics research in population-based studies with longitudinal assessment of dietary intake, using validated assessment tools, and serial collection of blood and urine, for generating metabolomics data, is lacking." This suggests a longitudinal analysis will be completed as part of this research, but don't think this is what the authors have done. I would argue averaging two or more time points into one then analysing is not a longitudinal analysis. Further methodological detail provided below. 

4. Methods

4.1 The IDATA study population and design, Dietary data collection, Nova classification for ultra-processed foods (UPF)

4.1.1 These sections are clear and well-articulated. 

4.1.2 I note that while participants may have provided up to six ASA-24s, the study entry criteria required only one. The authors have addressed this by averaging ASA-24-derived UPF percentages. This assumes that a patient with one ASA-24 (at a single point in time) is comparable to a patient with many more (over a longer period). If this is correct, then further justification of this approach is required. Also, averaging results from multiple timepoints removes much of the value of having longitudinal data. 

4.2 Biological sample collection, Metabolomics analysis, Quality Control (QC)

4.2.1 I am not a lab scientist so I cannot talk to the veracity of the science in these sections, but I did find them well-written and relatively understandable. 

4.2.2 [MINOR] Consider moving the QC results to the Results section.

4.2.3 With respect to the QC results, six-month temporal ICCs appear low suggesting differences in metabolite profiles over time. This finding suggests simply averaging metabolite profiles over time is potentially misleading e.g., high at one time point and low at a subsequent time point is assumed equivalent to medium at both times. 

4.2.4 Following on from this, was the possibility of using multiple pairings of biospecimens with ASA-24-derived UPF proportions instead of just averaging across the total follow-up time then analysing averages? 

4.3 Statistical analysis

4.3.1 I am unclear why a Kolmogorov-Smirnov test is required here. Please justify and ensure results are presented and discussed (or remove). 

4.3.2 Why wasn't alpha controlled for the multiple Kruskal-Wallis rank sum tests, noting 21 separate p-values presented in Table 2?

4.3.3 Why was a correlation-methods based approach used generally and why was partial Spearman rank correlation used specifically? I am not saying these are incorrect, rather these are not the only choices for these types of analyses (c.f., DOIs 10.1007/s11306-015-0823-6 and 10.1186/s12859-023-05383-0 ) and some justification should be provided. 

4.3.4 [MINOR] Suggest providing appropriate methodological citations for statistical methods including Benjamini-Hochberg method and LASSO regression.

4.3.5 The methodological detail for LASSO regression is good but some important things are not yet sufficiently clear. What precisely is the outcome being analysed - ordered quintiles like above or something else like actual proportion of energy derived from UPFs? Based on the outcome chosen, what statistical model is used to relate metabolites to UPFs? How are individual metabolites being operationalised - binary presence or absence based on a predetermined level? Actual metabolite levels? Put simply, what is the structure of the regression model that is used with LASSO to select candidate metabolites? Note this detail could be provided in the supplementary materials if word count is an issue. 

4.3.6 Relatedly, what exactly is the "poly-metabolite score"? I think it may be the predicted proportion of energy derived from UPFs? If this is correct, I would drop the use of "score" completely and just say "predicted proportion of UPF energy" or similar. If the score is something else, please provide detail on how it is calculated and justification regarding why it was chosen. Either way, ensure it is referred to logically in the rest of the manuscript. 

4.4 UPF poly-metabolite score applied to randomized, controlled, crossover-feeding trial

4.4.1 The use of paired sample t-test in this way is unfamiliar to me. Please provide justification for this choice with appropriate methodological citation(s). One alternative approach is to take the outcome to be predicted as binary (high vs. low UPF) and use an appropriate statistical model with the "UPF poly-metabolite score" as a fixed effect and accounting for paired samples. Also consider a more structured, formal approach to validation e.g., DOI 10.1093/eurheartj/ehu207

4.4.2 Either way, please articulate more clearly how you combined the analyses of the 20- and 4- patient groups. 

5. Results

5.1 Participant characteristics

5.1.1 [MINOR] "Average daily fiber density was also lowest (7.9 g/1,000 kcal) among those in the highest versus the lowest (10.7 g/1,000 kcal) quintile of average UPF intake (p-value<0.001)." Take care with wording here - this could be taken to be a p-value for the contrast comparing the lowest and highest quintiles, whereas I believe the p-value presented is for difference collectively across all quintiles. Consider this for all results when discussing constituent parts tested by an "overall" statistical test. 

5.2 Metabolite correlations with UPF intake

5.2.1 "(r ≥0.20 & r ≤-0.20, p.FDR ≤ 0.01)" is taken to be "strongly correlated" in several places. This seems to conflate two different things: strength of correlation (r) and strength of the evidence (p). Generally speaking, r = +/- .2 is not "strongly correlated" even with a small p-value. Consider aligning your wording regarding strength of correlation with generally accepted boundaries or justifying your thresholds. Suggest reflecting these definitions in the Methods. 

5.3 Pathway analysis

5.3.1 So these seven pathways each had combined p.FDR <= 0.01 for all three biospecimen types? Consider clarifying.

5.3.2 Why only consider pathways common across all biospecimen types where all other presented analyses treat biospecimen types separately? 

5.4 Poly-metabolite score generation using IDATA study

5.4.1 "… correlations between the scores and UPF intake were high (≥0.47) …" Assuming this is p≥0.47, similar to my previous point - I question whether 0.47 is "high" correlation. Suggest justification or rewording. 

5.5 UPF poly-metabolite score applied to a randomized, controlled, crossover-feeding trial

5.5.1 Aside from the concerns about the approach already raised, please ensure any presented statistics are described in the methods (noting ANOVA p-values presented in the supplementary results). 

6.0 Discussion and Conclusion

6.1 I will not repeat feedback already provided when similar feedback applies to the discussion, but please ensure any changes made in the earlier sections are reflected here too. 

6.2 "The positive correlation between N6-carboxymethylysine and UPF intake in our study suggests that a dietary pattern high in UPF may, in part, increase risk of developing type 2 diabetes (46) through this metabolite or related metabolic pathways." Correlation does not imply causation. Please reword. 

6.3 "Thus, our novel poly-metabolite scores could be used as an objective and complementary measure of UPF intake in future epidemiological investigations." Two things. Firstly, is this justified or is further validation of the presented "poly-metabolite score" appropriate first? Secondly, much is said in the introduction about combining self-report dietary information and biomarkers to improve accuracy and power (particularly using regression calibration), but this is not revisited in the discussion. 

6.4 "Firstly, diet over 12 months was assessed via multiple validated 24-hour dietary recalls, with more than 97% of participants completing four or more. This helped minimize measurement error due to random within-person day-to-day variation in dietary intake, and it also allowed for more accurate and detailed dietary data collection than a food frequency questionnaire." I question the veracity of this strength. As discussed previously, averaging away within patient variability does not necessarily better reflect reality compared with, say, modelling it directly with multiple time points per patient. 

6.5 The conclusion is succinct, well-justified, and clear. Well done! 

7.0 Tables and Figures

7.1 I will not repeat feedback already provided when similar feedback applies to the tables and figures, but please ensure any changes made in the earlier sections are reflected here too.

7.2. Some of the figures (main and supplementary) are very detailed and difficult to read. Consider splitting into separate sub-figures and/or ensure vector graphics (e.g., EMF or SVG in Windows) are provided, as they are sharp to arbitrary zoom levels. 

7.3 Suggest not using exponential notation for p-values

7.4 I am not sure what is going on with female vs. male in Figure S2. Please adjust so separation is clear. 

Reviewer #2: The manuscript submitted by Abar and colleagues describes the association of urine and serum metabolomic profiles with ultra-processed food intake in a population of 718 individuals. The manuscript is generally logically and descriptively written and in most places is easy to understand (though with some exceptions). The experimental design is generally appropriate though the population studied was very focused and further studies in different populations will be required to validate in the future. The conclusions derived are appropriate based on the data presented and it was nice to see the conclusions validated in an independent study where data was already available. There are some information and methods not included which should be included and although the strengths of the study were discussed, the weaknesses of the study should also be discussed.

There are a number of aspects which the authors should consider to provide a more descriptive and balanced manuscript.

1. Although supplementary file 1 describes the workflow for choice of participants it is somewhat confusing as to when the questionnaire was completed compared to when the biosamples were collected. This is important because there is an assumption that diet is static across the 12 months and so the biosamples represent the questionnaire-defined diet. A further description to detail this (and possibly a figure) would benefit the reader. Also, discussing this in the limtations would be appropriate.

2. Page 7. Blood samples were defined but which type of blood sample (serum or plasma), if plasma what was the anti-coagulant and is serum how was coagulation performed?

3. The metabolomics methods are poorly described and requiure significant more information.

4. Page 8. The metabolomic methods describe tandem mass spectrometry which suggests that a triple quadrupole mass spectrometer is applied and that data is reported as concentrations. I believe this is not the case for this service provider and so more details of the instrument model and manufacturer is required. Following on from this, more information on the types of liquid chromatography and methods applied is required.

5. Page 8. Osmolarity measurements were performed for urine. Why? No further mention of these data is included in the manuscript. These data would normally be used for pre- or post-analysis normalisation but this is not described.

6. Page 8 and elsewhere. Proof read the manuscript; for example, micro greek symbol and not u when defining microlitres.

7. Page 8. Fixed concentrations of standards. What are these standards and how are they applied? I assume they are isotopically-labelled internal standards and if so how is the data used to assess quality of the data or report concentrations.

8. Page 8. How are metabolites identified? MS1, MS/MS, RT, other?

9. Page 9. Normalised by run day. What does this mean as it is not something commonly applied in metabolomics. Also how are data normalised and scaled for data analysis?

10. Page 11. What definition of normal weight, overweight and obese was used as this can differ between continents?

11. Page 12 onwards. The authors describe some metabolites as amino acids which in reality are not amino acids and will confuse some readers. The authors should define clearly how these metabolites are classified, i.e. assume based on Metabolon classification system. Some point for 4-deoxythreonic acid which is not a lipid.

12. Page 12. C18H21NO3, all numbers should be subscripted to be correct.

13. Page 14, pathway analysis. The methods applied for pathway analysis should be included in the methods section.

14. The discussion should discuss the origin of the important metabolites. Are these diet-derived, gut microbiome-derived or human responses to dietary components.

15. Figure 1. These heatmaps show correlations between metabolites and other metabolites as well as UPF intake. This makes these figures difficult to view and understand especially with the small size of the text. It would be more appropriate to visualise the corrections between metabolites and UPF food intake only.

16. Supplementary figure 1 includes errors, some text is cut-off and not visible and u used for micro.

17. The manuscript discussed strengths of the study but not weaknesses. To provide a balanced story both strengths and weaknesses should be discussed.

(i) The population sampled is not diverse in age (all >50 years) and probably educated as access to internet required as well as being predominantly white and non-hispanic.

(ii) The biosamples were collected at different times to the questionnaire data and so there is an assumption that the diet at the time of the questionnaire is the same as when biosamples were collected. As metabolomic profiles are dynamic, if diet changes this will be viewed in the profiles quickly within hours normally.

(iii) It is unclear which types of metabolites and lipids were studied. The results predominantly discuss non-lipids and it is unclear whether this is because lipids were not studied or because lipids were studied but where not statistically significant. You would expect lipids to change significantly with UPF.

Reviewer #3: The manuscript by Abar et al. describes the metabolomic investigation of long-term ultra-processed food intake in the longitudinal cohort called Interactive Diet and Activity Tracking in AARP (IDATA) Study. This large cohort study investigated urine and serum samples of 787 individuals using the Metabolon platform. The authors identified metabolic predictors for the ultra processed foods (UPFs) using a LASSO regression model, which can be used for determining UPF dietary intake. The study has been performed well and the manuscript is well written. The authors have just missed to discuss their powerful large dataset in more detail, which will improve the impact of their important study covering the medical and nutritional field. In summary, I recommend publication of this study in PLoS Medicine after my remarks have been addressed.

Major remarks:

- The identified metabolites seem to have a high predictive power. To compare the value of their biomarker pattern with other studies, it is important to perform receiver operating characteristics analysis.

- An overview figure to describe the study design of this study and the cohort would be benefit the manuscript.

- The authors describe pathway analysis throughout the manuscript but the discussion on these pathway is very brief. I would like to see a more thorough description of the identified pathways, how they have been linked previously to dietary differences and what can be medically learned from these alterations.

- This sentence should be moved or summarized in the conclusion section: "Thus, the metabolomic profiles that we identified likely reflect a dietary pattern characterized not only by high UPF consumption but also by low whole food, including fresh fruits and vegetables, consumption."

- Figure 1: All text should have the size of the left panel in the first top left heatmap.

- The resolution of Figure 3 must be improved.

Minor remarks:

- "meat eater" is a trivial word and should be replaced by the correct scientific term for this cohort.

- Instersection should be correct to Intersection in the Figure 2 legend.

Reviewer #4: PMEDICINE-D-24-02724R1

Serum and urine metabolomic profiles of long-term ultra-processed food intake in the 

longitudinal Interactive Diet and Activity Tracking in AARP (IDATA) Study

In this study, Abar and colleagues tried to discover intake biomarkers of ultra-processed food in the participants' serum and urine samples and replicated the the intake prediction based on such panel of discovered biomarkers in an independent set of samples from 20 participants of an intervention study. While finding a set of biomarkers for ultra-processed foods is important, there is something the authors need to clarify. Previous studies usually used randomized, controlled trial (RCT). as the amount of intake is fully controlled and free from any reporting or recall bias commonly occurred in a cohort study (for example, Willis et al 2020 Front Nutr doi 10.3389/fnut.2020.561010). The replication could then be done in a free-living cohort study with habitual diets (as suggested by Maruvada et al 2020 Adv Nutr doi 10.1093/advances/nmz075). Another option is to split the original cohort to a training and a test set, where the training set was used to discover the set of biomarkers, and then the predicted intake was tested against an independent subset of the same cohort (test set, for example, Li et al, 2022 AJCN doi 10.1093/ajcn/nqac094). The authors, however, used the large cohort as a discovery study with > 700 participants and test the derived biomarkers in a much smaller RCT with 20 participants. While there is no strict protocol on how to derive a set of intake biomarkers, did the authors have any special reason to use such a not-so-common "reversed" approach? And could this be also the reason for the observed variability (page 15, Supplementary Table 7)? This selection of methods needs to be justified either in the methods or in the discussion section.

---

* Please upload any figures associated with your paper as individual TIF or EPS files with 300dpi resolution at resubmission; please read our figure guidelines for more information on our requirements: http://journals.plos.org/plosmedicine/s/figures. While revising your submission, please upload your figure files to the PACE digital diagnostic tool, https://pacev2.apexcovantage.com/. PACE helps ensure that figures meet PLOS requirements. To use PACE, you must first register as a user. Then, login and navigate to the UPLOAD tab, where you will find detailed instructions on how to use the tool. If you encounter any issues or have any questions when using PACE, please email us at PLOSMedicine@plos.org.

* PLOS Medicine requires that the de-identified data underlying the specific results in a published article be made available, without restrictions on access, in a public repository or as Supporting Information at the time of article publication, provided it is legal and ethical to do so. Please see the policy at 

http://journals.plos.org/plosmedicine/s/data-availability

and FAQs at 

http://journals.plos.org/plosmedicine/s/data-availability#loc-faqs-for-data-policy

* Please outline why your manuscript does not require ethics approval

FIGURES AND TABLES

SUPPLEMENTARY MATERIAL

REFERENCES

RCTs 

* PLOS Medicine requires that all trials be prospectively registered in one of registries recognized by WHO. Please ensure that study registration details are included in the Methods section.

* Please structure the Methods section using the following sub-headings: Study design and participants, Randomization and masking, Procedures, Outcomes, Statistical analysis.

* Please ensure that all prespecified outcomes (primary, secondary, and exploratory) are listed in the Methods/Outcomes section and indicate whether there are outcomes that are not presented in the current report.

* Please specify the dates (Month Day, Year) during which study enrollment and follow up occurred.

* Please include absolute numbers wherever you report percentages; eg, n/N (%)

* Please present the safety data for the study including numbers of specific events and whether or not adverse events are thought to be related to treatment. AEs should be reported in the abstract, per CONSORT and CONSORT-Harms.

* Please complete the CONSORT checklist (https://www.equator-network.org/reporting-guidelines/consort/) and ensure that all components of CONSORT are present in the manuscript, including how randomization was performed, allocation concealment, blinding of intervention, definition of lost to follow-up, power statement. When completing the checklist, please use section and paragraph numbers, rather than page numbers.

* Please report your abstract according to CONSORT for abstracts, following the PLOS Medicine abstract structure (Background, Methods and Findings, Conclusions) https://www.equator-network.org/reporting-guidelines/consort-abstracts/

* If your trial had to undergo important modifications in response to extenuating circumstances, please complete the CONSERVE-CONSORT checklist and provide in your Supporting Information; (https://www.equator-network.org/reporting-guidelines/guidelines-for-reporting-trial-protocols-and-completed-trials-modified-due-to-the-covid-19-pandemic-and-other-extenuating-circumstances-the-conserve-2021-statement/). When completing the checklist, please use section and paragraph numbers, rather than page numbers.

* In keeping with our commitment to Open Science, please include the study protocol document and analysis plan (including any amendments) as Supporting Information to be published with the manuscript if accepted.

* Please note that PLOS Medicine requires prospective, public registration of a data sharing plan (as part of mandatory clinical trials registration) for all clinical trials that began enrollment on or after January 1, 2019, in accordance with ICMJE requirements.

---

## [Decision Letter · Decision Letter 2]

17 Mar 2025

Dear Dr. Abar,

Thank you very much for re-submitting your manuscript "Identification of Poly-Metabolite Scores for Diets High in Ultra-Processed Food in an Observational Study with Validation in a Randomized Controlled Crossover-Feeding Trial" (PMEDICINE-D-24-02724R2) for review by PLOS Medicine.

I have discussed the paper with my colleagues and it was also seen again by three reviewers. I am pleased to say that provided the remaining reviewer concerns and the editorial and production issues are dealt with we are planning to accept the paper for publication in the journal. However, we do plan to request feedback from the editor and reviewers on the revised manuscript before making any final decisions on the manuscript.

[LINK]

We look forward to receiving the revised manuscript by Mar 24 2025 11:59PM.   

Sincerely,

Rebecca Kirk, 

Senior Editor 

PLOS Medicine

plosmedicine.org

Requests from Editors:

GENERAL EDITORIAL REQUESTS

* At this stage, we ask that you include a short, non-technical Author Summary of your research to make findings accessible to a wide audience that includes both scientists and non-scientists. The Author Summary should immediately follow the Abstract in your revised manuscript. This text is subject to editorial change and should be distinct from the scientific abstract. Ideally each sub-heading should contain 2-3 single sentence, concise bullet points containing the most salient points from your study. In the final bullet point of ‘What Do These Findings Mean?’ Please include the main limitations of the study in non-technical language.

Please see our author guidelines for more information: https://journals.plos.org/plosmedicine/s/revising-your-manuscript#loc-author-summary."

* Please confirm that your title complies with to PLOS Medicine's style. Your title must be nondeclarative and not a question. It should begin with main concept if possible. "Effect of" should be used only if causality can be inferred, i.e., for an RCT. Please place the study design ("A randomized controlled trial," "A retrospective study," "A modelling study," etc.) in the subtitle (ie, after a colon).

* Please confirm that your abstract complies with our requirements, including providing all the information relevant to this study type https://journals.plos.org/plosmedicine/s/submission-guidelines#loc-abstract

* Please ensure that the Introduction ends with a clear description of the study question or hypothesis.

* Please ensure that all abbreviations are defined at first use throughout the text.

* Please remove the 'conclusions' subheading.

FUNDING STATEMENT

* The funding statement should include: specific grant numbers, initials of authors who received each award, URLs to sponsors’ websites. Also, please state whether any sponsors or funders (other than the named authors) played any role in study design, data collection and analysis, the decision to publish, or preparation of the manuscript. If they had no role in the research, include this sentence: “The funders had no role in study design, data collection and analysis, decision to publish, or preparation of the manuscript.”

COMPETING INTERESTS STATEMENT

* All authors must declare their relevant competing interests per the PLOS policy, which can be seen here: https://journals.plos.org/plosmedicine/s/competing-interests For authors with ties to industry, please indicate whether any of the interests has a financial stake in the results of the current study.

DATA AVAILABILITY

* The Data Availability Statement (DAS) requires revision and the access information should be included in the main text of the manuscript, not only in the Supplementary Information. For each data source used in your study: 

FIGURES

* Please define all elements of box plots in the figure caption - center line, box limits and whiskers.

* Please convert any stacked bar charts to another data representation for example a table, or other type of graph.

CLINICAL TRIALS

* Please complete the CONSORT checklist and ensure that all components of CONSORT are present in the manuscript, including how randomization was performed, allocation concealment, blinding of intervention, definition of lost to follow-up, power statement. When completing the checklist, please use section and paragraph numbers, rather than page numbers. The checklist should be included as supporting information, and should be cited in the article.

* The CONSORT flowchart should be figure 1, please revise.

* Please report your abstract according to CONSORT for abstracts, following the PLOS Medicine abstract structure (Background, Methods and Findings, Conclusions) https://www.equator-network.org/reporting-guidelines/consort-abstracts/

* Please include the clinical trial registry number in the abstract.

Comments from Reviewers:

Reviewer #1: Thanks to the authors for the informative and comprehensive responses to my feedback and associated changes to the manuscript. In all cases I agree and consider all items closed. The only very minor suggestion I have relates to 3.2. I suggest you cite reference 36 (Sampson, J.N., et al.) in the added text at page 11, lines 233-236. This is important because it provides methodological information and justification for the adopted approach. 

That said, I am happy to recommend acceptance as-is. Well done and congratulations.

Reviewer #3: The author have improved their manuscript in this revision. However, the claim that the ROC curves can be include in a future manuscript is incorrect. The authors should include this analysis for researchers in the field to compare the value of their molecules to other molecules identified in other studies for Ultra-Processed Food. Otherwise, I cannot recommend publication of this manuscript in PLoS Medicine.

[LINK]

---

## [Decision Letter · Decision Letter 3]

3 Apr 2025

Dear Dr Loftfield, 

On behalf of my colleagues and the Academic Editor, Fumiaki Imamura, I am pleased to inform you that subject to the requirements below we have agreed to publish your manuscript "Identification and Validation of Poly-Metabolite Scores for Diets High in Ultra-Processed Food: An Observational Study and a Randomized Controlled Crossover-Feeding Trial" (PMEDICINE-D-24-02724R3) in PLOS Medicine.

There are some additional requirements that will need to be addressed:

* The metabolomics data should be made fully available for reuse, one recommended repository for these data is available at MetaboLights - https://www.ebi.ac.uk/metabolights/ Access to the dataset should not be limited to a single contact email address.

* Please provide an ethics statement in the document to include the details of the IRB approval number referred to on page 6. Please ensure that the approval details for both studies are available, and in a clearly defined ethics section.

* You have confirmed that the use of the trial data is post hoc in your rebuttal, but this is only referred to in the Results section in the manuscript. Please make this clearer in the Abstract and Methods, and please update the title to reflect this as well. Suggestion: Poly-Metabolite Scores for Diets High in Ultra-Processed Foods: An Observational Study and Post-Hoc Analysis of a Randomized Controlled Crossover Feeding Trial

PRESS

Sincerely, 

Rebecca

Rebecca Kirk 

Senior Editor 

PLOS Medicine